Methods

# An immune-based tool platform for in vivo cell clearance

Jieqiong Zhang[1], Tatsuya Tsukui[2], Xiumin Wu[1], Alyssa Brito[3], John Maxwell Trumble[3], Juan C Caraballo[2], Greg M Allen[4], José Zavala-Solorio[1], Chunlian Zhang[1], Jonathan Paw[1], Wendell A Lim[4,5], Jiefei Geng[3], Yuliya Kutskova[3], Adam Freund[1], Ganesh Kolumam[1], Dean Sheppard[2], Robert L Cohen[1]

Immunological targeting of pathological cells has been successful in oncology and is expanding to other pathobiological contexts. Here, we present a flexible platform that allows labeling cells of interest with the surface-expressed model antigen ovalbumin (OVA), which can be eliminated via either antigen-specific T cells or newly developed OVA antibodies. We demonstrate that hepatocytes can be effectively targeted by either modality. In contrast, pro-fibrotic fibroblasts associated with pulmonary fibrosis are only eliminated by T cells in initial experiments, which reduced collagen deposition in a fibrosis model. This new experimental platform will facilitate development of immune-based approaches to clear potential pathological cell types in vivo.

## Introduction

Pathological cells are the driving force of certain diseases, and their depletion serves as an attractive therapeutic strategy in many settings. The most prominent examples lie in cancer. The main focus of oncology aims to eliminate tumor cells to cure or slow down cancer progression. Besides tumor cells per se, other cancer-associated cell types are also being explored. Cancer-associated fibroblasts, despite their heterogeneity, are increasingly viewed as a target that can be manipulated for therapeutic benefit in cancer patients (Sahai et al, 2020). Myeloid-derived suppressor cells are a diverse population of immature myeloid cells that have potent immune–suppressive activity, and depletion of myeloid-derived suppressor cells is one strategy currently being explored to overcome immune evasion in cancer immunotherapy (De Cicco et al, 2020). Another increasingly well-established type of pathological cells is senescent cells. Senescent cells accumulate during aging and in age-associated diseases, and senolytics are actively being developed as anti-aging drugs (Di Micco et al, 2021). In addition, recent advances in single cell RNA-seq have also revealed novel disease-associated cell types at an unprecedented speed. However, it remains largely unexplored whether these cells cause disease progression and whether these potential pathological cells are good therapeutic targets.

Immune-based approaches are powerful in targeting pathological cells. First, they are specific, and have already demonstrated to be successful, or even revolutionary, in oncology. For example, antibody-based drugs that lead to cell lysis are now routinely used in oncology (Zahavi & Weiner, 2020), and so far, six chimeric antigen receptor T cell therapeutics (CAR-Ts) targeting CD19/BCMA have been approved by the FDA in treating blood cancers. Second, recent efforts have established that CAR-Ts can be successfully applied to target non-cancer cells in animal models. For example, targeting FAP+ myofibroblasts via traditional CAR-Ts reversed cardiac fibrosis (Aghajanian et al, 2019). Transient CAR-Ts produced in vivo via mRNA/LNP delivery had a similar effect (Rurik et al, 2022). Elimination of uPAR+ senescent cells via CAR-Ts also reduced liver fibrosis in vivo (Amor et al, 2020). To enable immune-based interventions aimed at a particular cell type, it is essential to first identify specific surface markers that could serve as good translational targets, followed by the generation of a clearing reagent, such as a therapeutic antibody, or CAR-T cells. These are challenging and resource-consuming efforts, which often hinder effective exploration of the pathobiological contribution of multiple cell types at scale.

So far, most in vivo studies aiming to eliminate cells use cell autonomous ablation mechanisms. In the most widely used mouse model based on diphtheria toxin receptor, diphtheria toxin triggers cell death of mouse cells that overexpress the engineered human receptor (Buch et al, 2005; Voehringer et al, 2008). Inclusion of suicide genes, like caspase 8 and thymidine kinase, has also been used to target senescent cells (Baker et al, 2011; Demaria et al, 2014). In contrast, immune-based tools for cell clearance, which better mimic the actual therapeutic modality, are much more limited. One idea that has been proven to be useful is to use antigen-specific T cells to target cells of interest with a model antigen, like GFP (Agudo et al, 2015) or OVA (Sandhu et al, 2011; Cebula et al, 2013).

We therefore decided to create an optimized platform that allows basic exploration of targeting different cell types with

---

[1]Calico Life Sciences LLC, South San Francisco, CA, USA [2]Division of Pulmonary, Critical Care, Allergy and Sleep, Department of Medicine, University of California, San Francisco, San Francisco, CA, USA [3]AbbVie Inc, Chicago IL, USA [4]Department of Medicine, University of California San Francisco, San Francisco, CA, USA [5]Cell Design Institute, University of California San Francisco, San Francisco, CA, USA

Correspondence: zjq.thu@gmail.com; rlc@calicolabs.com

 

immune-based approaches, and to evaluate its therapeutic potential. To this end, we generated a Cre reporter line in a C57BL/6 background that allows labeling cells of interest with a surface-expressed common antigen, OVA, and new OVA antibodies that can mediate cell killing. This approach is versatile as it allows labeling cells of interest based on the Cre-LoxP system. The platform also takes advantage of the existing antigen-specific OT-1 T cells (Hogquist et al, 1994) that can remove OVA+ cells in vivo. We further applied this system to target a group of fibrosis-associated fibroblasts, and showed that OT-1 T cells, but not the OVA antibody, eliminated those fibroblasts and reduced collagen production. The platform could be a powerful tool that allows systematic exploration of a variety of disease-associated cells as potential therapeutic targets via immune-based approaches.

# Results

## Generation and validation of the *Rosa26$^{LSL-OVA-Luc/+}$* mouse model

To provide a platform that enables immune-based clearance of specific cell types, we took advantage of the fact that OT-1 T cells are well-established antigen-specific T cells that can remove OVA+ cells in vivo. The specific labeling strategy is inspired by a well-established RIP-mOVA model (Kurts et al, 1996; Harbers et al, 2007), where the rat insulin promoter drives the expression of the transmembrane domain of the transferrin receptor fused-ovalbumin (TFRC-OVA) in β cells in pancreatic islets. Consistent with published reports, adoptive transfer of OT-1 T cells, but not WT T cells, resulted in diabetes in RIP-mOVA mice (Fig S1A). Histology analysis also revealed infiltration of CD8 and CD4 immune cells, and complete elimination of OVA+ cells in the islets upon administration of OT-1 T cells (Fig S1B).

Motivated by these results, we generated a new mouse model *Rosa26$^{LSL-OVA-Luc/+}$*, where a universal CAG promoter drives the expression of a LoxP-stop-LoxP cassette, the fusion gene TFRC-OVA and a luciferase reporter (Fig 1A).

We then verified the *Rosa26$^{LSL-OVA-Luc/+}$* model in vivo by crossing it with established Cre lines. For example, *Rosa26$^{LSL-OVA-Luc/+}$*, *actin-Cre* exhibited ubiquitous luciferase signals (Fig 1B), and widespread OVA expression in multiple tissues (Fig S2). In contrast, in *Rosa26$^{LSL-OVA-Luc/+}$*, *Alb-Cre* mice, luciferase signal, and OVA expression were restricted to liver, where Cre is expressed (Fig 1B and C). *Rosa26$^{LSL-OVA-Luc/+}$* mice also exhibited detectable basal level of luciferase signals (Fig S1C), probably because of the leaky expression of the LoxP-stop-LoxP cassette in the absence of Cre, although such basal level expression of OVA was barely detected in most tissues via immunohistochemistry analysis (Figs 1C and S2, middle panel). Together, these data established that the new *Rosa26$^{LSL-OVA-Luc/+}$* line serves as a Cre reporter line that allows labeling cells of interest with OVA, based on the Cre-LoxP system.

## Adoptive transfer of OT-1 T cells leads to liver damage in OVA+ liver model

The main advantage for labeling target cells with the model antigen OVA is to use existing tools (OT-1 T cells) for specific cell clearance. This is confirmed in vitro, using mouse embryonic fibroblasts (MEFs)

derived from *Rosa26$^{LSL-OVA-Luc/+}$*. These MEFs showed luciferase signals when infected with Cre lentivirus (Fig S1D), but not GFP (Fig 1D, first column, T cells/MEF ratio = 0). When the OVA+ MEFs were co-cultured with OT-1 T cells, they were specifically killed at a high E:T (effector: target) ratio, evidenced by the reduction of the luciferase signal (Fig 1D). Consistent with the in vitro validation results, adoptive transfer of OT-1 T cells, but not WT T cells led to elevated liver enzyme levels in the serum of *Rosa26$^{LSL-OVA-Luc/+}$*, *Alb-Cre* mice (Fig 1E), indicative of parenchymal liver damage when OVA+ hepatocytes were attacked by OT-1 T cells. Histology analysis also showed necrosis and inflammation in the damaged liver (Fig 1F). However, the reduction of the luciferase signal was not observed (Fig S1E), suggesting that only a small minority of liver cells was killed, potentially because of a low E:T ratio achieved in vivo for OVA+ hepatocytes. Collectively, these results showed that OT-1 T cell-based adoptive transfer can be used to eliminate OVA+ cells in mice based on the *Rosa26$^{LSL-OVA-Luc/+}$* model.

## Development of OVA antibodies to target OVA+ cells in vivo

In addition to the T cell-based approach, we also developed OVA antibodies as another strategy to target OVA+ cells. This strategy is possible as fusion to transferrin receptor allows OVA to be stably expressed at the cell surface, accessible to antibodies. We made two anti-OVA antibodies with the same variable region, but with different constant regions: OVA-mIgG2a and OVA-mIgG1. The active mIgG2a isotype, but not mIgG1, binds to CD16-2 (mFcγRIV) normally present on macrophages, leading to lysis of the target cells via antibody-dependent cellular cytotoxicity (ADCC).

To evaluate the OVA antibodies, single-cell clones of NIH3T3 cells overexpressing different levels of OVA were established as targeting cells (Fig S3A). Although both antibodies bind to these OVA+ cells with similar affinity (Fig S3B), only the active OVA-mIgG2a antibody exhibited activity in an ADCC reporter assay (Fig 2A). The OVA-mIgG2a antibody was more effective in OVA$^{high}$ than in OVA$^{low}$ cells, and showed no activity in the NIH3T3 parental cells. The mIgG1 isotype control did not induce ADCC activity in either parental or OVA+ cells. These results suggested that the antibody is specific, and its ADCC activity is dependent on the expression level of OVA in the targeting cells.

We next tested whether the OVA antibodies also exhibited activity in vivo. *Rosa26$^{LSL-OVA-Luc/+}$*, *Alb-Cre* mice that were injected intraperitoneally with OVA-mIgG2a antibodies had increased serum liver enzyme levels with liver granulomas and inflammation by histology, whereas mice injected with OVA-mIgG1 antibodies did not (Fig 2B and C). Antibody-induced killing is specific, as liver enzyme levels in the RIP-mOVA (OVA+ β cell) model remained normal (Fig S3C). However, in the β cell model, neither antibody-based elimination of OVA+ β cells nor elevated serum glucose was observed after 2 wk of antibody dosing (Fig S3D and E). Together, these results show that the ADCC activity of the OVA antibody is influenced by the OVA expression level in target cells, resulting in context-dependent cytotoxicity.

## Elimination of *Cthrc1*-OVA+ cells by OT-1 T cells leads to mild reduction of fibrosis in vivo

Next, we applied the OVA platform in a disease model. Here, we focus on *Cthrc1+* fibroblasts, a recently identified sub-type that is

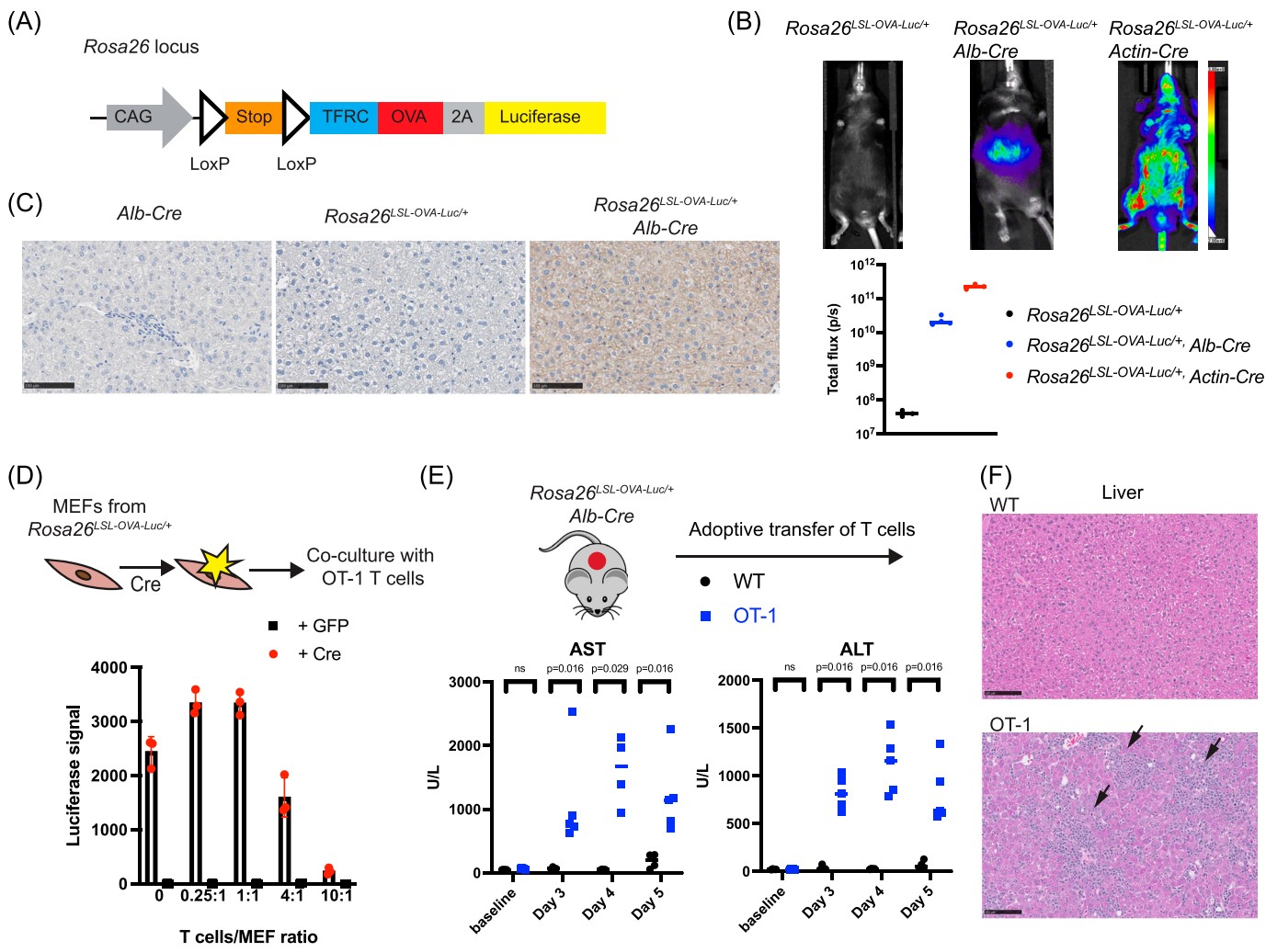

**Figure 1. Generation and validation of the *Rosa26^LSL-OVA-Luc* mouse model.**
**(A)** Schematic representation of the knock-in allele (CAG promoter driving the LoxP-stop-LoxP-TFRC-OVA-Luciferase cassette) at the *Rosa26* locus. **(B)** Representative luciferase imaging results of *Rosa26^LSL-OVA-Luc/+*, crossed with different Cre lines. Quantification of the results are shown in the panel below. n = 3 or 4 mice per genotype. Symbols represent individual mice. Lines indicate the median. **(C)** Immunohistochemical (IHC) labeling for OVA (n = 3 mice per genotype). Scale bar, 100 μm. **(D)** Co-culture assays of MEFs and OT-1 T cells. MEFs derived from *Rosa26^LSL-OVA-Luc/+* mice were infected with either GFP or Cre virus, and co-cultured with pre-activated OT-1 T cells. Luciferase signal was measured 24 h after the co-culture. Symbols represent individual mice. Lines indicate the mean with SD. **(E)** Serum liver enzyme analysis post adoptive transfer of T cells in *Rosa26^LSL-OVA-Luc/+*, *Alb-Cre* mice. Symbols represent individual mice. Lines indicate the median. Multiple Mann–Whitney tests were used to determine *P*-values. **(F)** Representative images of histologic analysis of the liver. Mice from (E) were taken down at Day 5 post adoptive transfer. n = 4 for mice transferred with WT T cells; n = 5 for mice transferred with OT-1 T cells. Scale bar, 100 μm. Arrowheads, inflammation, indicated by the infiltration of immune cells.

not present in normal conditions, but which accumulates at sites of active fibrogenesis in the bleomycin-induced lung fibrosis model and in idiopathic pulmonary fibrosis patients (Tsukui et al, 2020). Depletion of *Cthrc1+* cells via a diphtheria toxin-based model showed mild reduction of collagen composition (Tsukui & Sheppard, 2022 *Preprint*), suggesting that these fibroblasts could be a potential target for therapeutic modulation. However, it is unclear whether immune-based approaches are feasible ways to efficiently target *Cthrc1+* cells, and if so, whether they also offer therapeutic benefits.

We therefore crossed *Cthrc1-CreERT2* to *Rosa26^Td-tomato/+* and *Rosa26^LSL-OVA-Luc/+* so that *Cthrc1+* cells, upon tamoxifen induction,

can be permanently labeled as tdTomato+ and OVA+ cells. As expected, bleomycin induced tdTomato+ cells, as shown by microscopy and FACS analysis (Figs 3A–C and S4A [Tsukui & Sheppard, 2022 *Preprint*]). Adoptive transfer of GFP-labeled OT-1 T cells resulted in infiltration of GFP+ OT-1 T cells and significant reduction in the number of tdTomato+ *Cthrc1+* cells (Figs 3B and C and S4A). qRT-PCR analysis for whole-lung cells also confirmed strong reduction of *Cthrc1* and *Col1a1* (Collagen 1a1) expression (Fig 3D), further supporting the efficient removal of the *Cthrc1+* profibrotic cells, the main cell types contributing to the expression of fibrotic genes (Tsukui et al, 2020). In contrast, administration of the OVA antibodies for 2 wk did not lead to reduction of tdTomato+ *Cthrc1+*

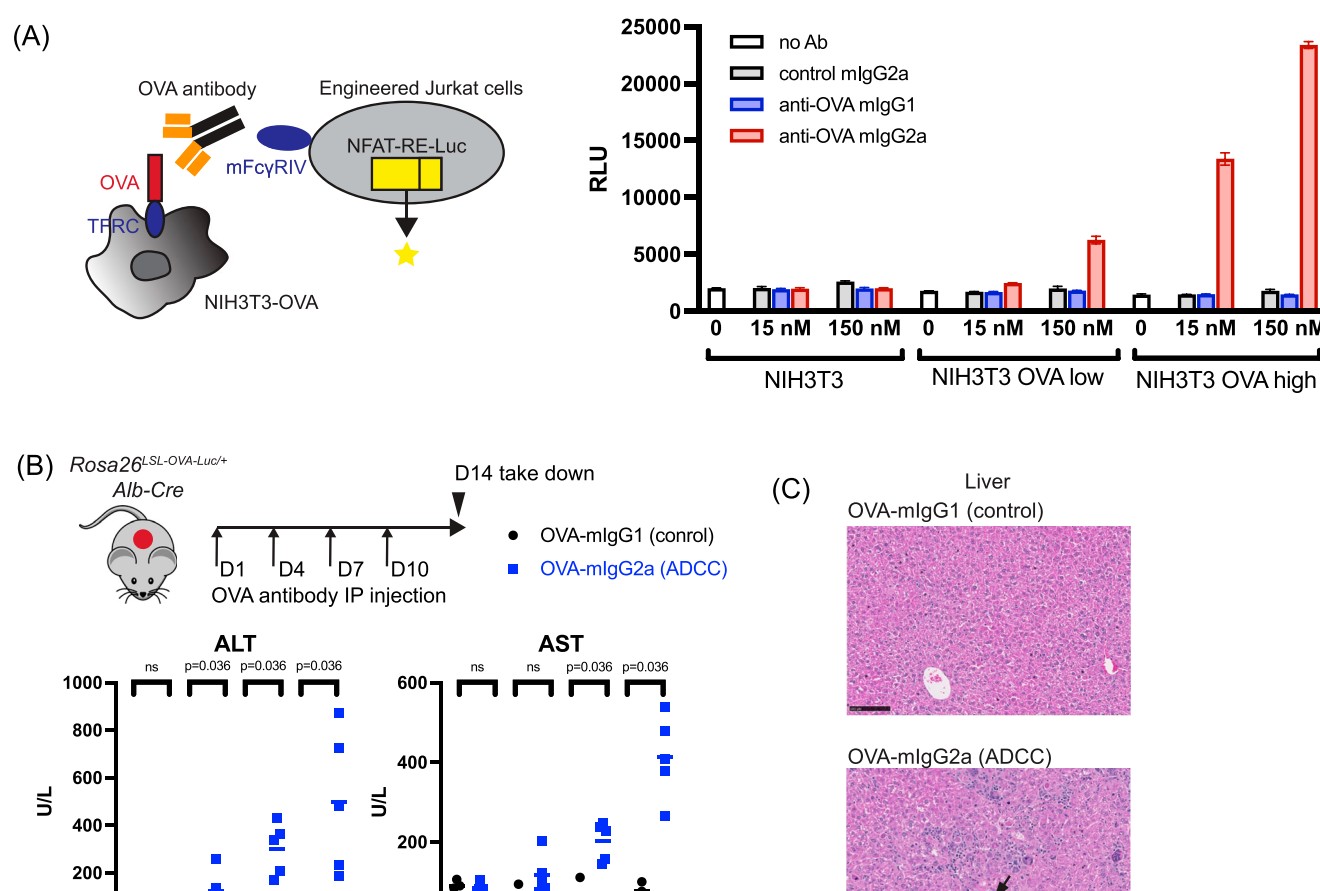

**Figure 2. Development and validation of OVA antibodies for cell clearance.**
**(A)** Left panel: cartoon illustrating the OVA antibody-dependent ADCC assay. Right panel: NIH3T3 cells overexpressing different levels of OVA were co-cultured with the engineered Jurkat cells in the presence of different antibodies at indicated concentrations. Nano-luciferase activity was measured after 6 h co-culture. Triplicate results were shown. Lines indicate the mean with SD. **(B)** Serum liver enzyme analysis post antibody injection in *Rosa26$^{LSL-OVA-Luc/+}$*, *Alb-Cre* mice. Symbols represent individual mice. Lines indicate mean. Multiple Mann–Whitney tests were used to determine *P*-values. **(C)** H&E staining of liver histology analysis. Mice from (B) were taken down at Day 14. n = 3 for mice injected with mIgG1 control antibody; n = 5 for mice injected with mIgG2a antibody. Scale bar, 100 *μ*m. Arrowheads: inflammation and granuloma.

cells (Fig S4B–D). This set of experiments showed that the OVA system can be applied to specifically label a disease-associated cell type. In this fibrosis model, T cells rather than the OVA antibody are much more effective in eliminating *Cthrc1+* cells in vivo.

Importantly, in mice where *Cthrc1+* fibroblasts were eliminated, fibrosis measured by the hydroxyproline content was also modestly reduced (Fig 3E). Together, these results suggest although the elimination of *Cthrc1+* cells can be effective, the therapeutic benefits in improving fibrosis are relatively mild. It is likely that our approach does not inhibit early fibrogenesis, which happens even before the T cells were introduced (Day 14). Therefore, the marginal therapeutic benefits are not too surprising given the complicated factors contributing to fibrosis and, of course, an immune-mediated cytotoxic approach is just one therapeutic strategy.

## Discussion

Here, we present a new platform that enables labeling cells of interest with a membrane-bound version of OVA, based on the newly generated *Rosa26$^{LSL-OVA-Luc/+}$* model. Whereas T cells are effective in multiple contexts, the OVA ADCC antibody is effective only in the liver model, probably reflecting differences in the expression level of surface protein and the penetrance of antibodies in the local tissue environment. These tools therefore can be used to evaluate whether disease-associated cell types can be a good therapeutic target via different immune-based approaches.

Although earlier models based on cell-autonomous killing cassettes provide powerful approaches to "knockout" cells in vivo, the OVA platform can provide more translational insights

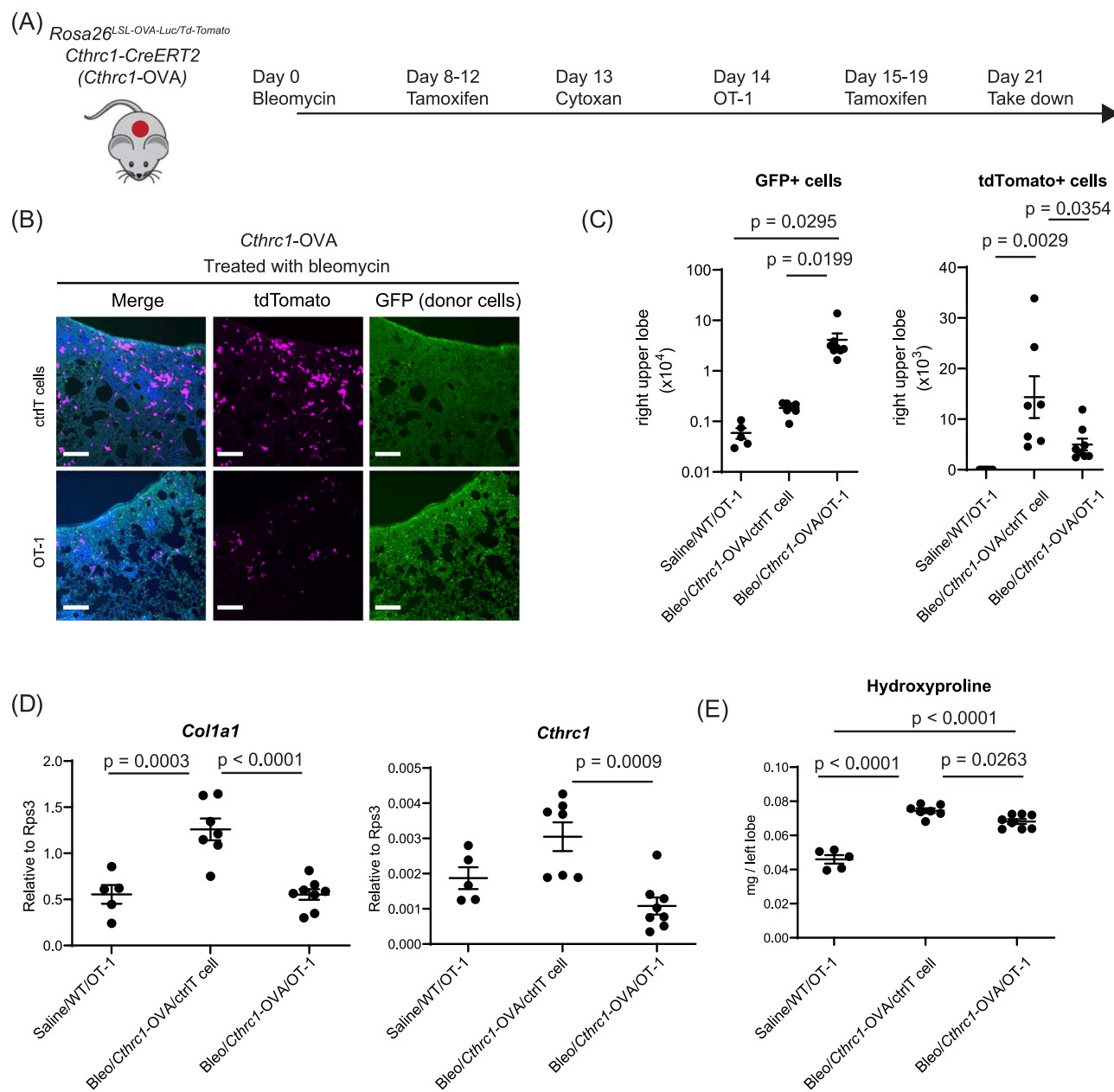

**Figure 3. Elimination of *Cthrc1*-OVA+ cells by OT-1 T cells leads to mild reduction of fibrosis in vivo.**
**(A)** Cartoon illustrating the experimental design. **(B)** Lung sections of *Rosa26^td-Tomato/LSL-OVA-Luc*, *Cthrc1-CreERT2* treated with bleomycin, showing tdTomato+ *Cthrc1* pro-fibrotic fibroblasts (magenta) and GFP+ transferred control or OT-1 T cells (green). DAPI staining is shown in blue. Scale bars, 200 μm. Data are representative from n = 7 mice transferred with control T cells and n = 7 for mice transferred with OT-1 T cells. **(C)** Cell numbers of GFP+ cells (left) or tdTomato+ cells (right) in the right upper lobe counted by FACS. Lines indicate mean ± SEM. One-way ANOVA followed by the Tukey–Kramer post-test was used to assess *P*-values. **(D)** qRT-PCR analysis of whole lung cells for *Col1a1* and *Cthrc1*. Y axis is the relative expression level to a house-keeping gene *Rps3*. Lines indicate mean ± SEM. One-way ANOVA followed by the Tukey–Kramer post-test was used to assess *P*-values. **(E)** Hydroxyproline assay showing reduced fibrosis by deletion of pro-fibrotic fibroblasts by OT-1 transfer. Data are mean ± SEM. One-way ANOVA followed by the Tukey–Kramer post-test was used to assess *P*-values.

extending beyond the biological consequences of cell ablation. For example, the OVA mouse model can be an early tool for characterizing the safety profile, potential disease indications, and preferred therapeutic approaches (antibody versus T cells). It can also serve as a good preclinical model to test if immuno-oncology drugs can be further extended in non-oncology indications. In the context of *Cthrc1+* pro-fibrotic fibroblasts, it will be interesting to test whether immuno-oncology drugs that boost the T cell killing efficiency against tumors also help to eliminate *Cthrc1+* cells, potentially leading to better therapeutic benefits in fibrosis. Specifically, the *Cthrc1*-OVA model will allow us to test the combination of OT-1 T cells/OVA antibodies together with potential immune-boosters in the fibrosis model.

More generally, the OVA platform is a pilot tool to explore immune-based approaches in mouse models of disease, without prior knowledge of known surface markers, an essential requirement for developing therapeutic CAR-Ts or antibodies. Notably, CTHRC1 itself is a soluble protein. By utilizing the Cre-LoxP system, *Ctrhc1+* cells are turned into OVA+ cells, making it possible to evaluate the effects of cell killing using OT-1 T cells and OVA antibodies. One important caveat of the system is that OVA is exogenously introduced and is typically overexpressed at an artificially high level. Therefore, having established that a cell type is worth pursuing, it will be important to take into consideration the immunogenicity, specificity, endogenous copy number of the candidate surface markers, and number of pathological cells that need to be targeted. In terms of *CTHRC1+* cells, one candidate of particular interest is *LRRC15*. Single-cell analysis of fibroblast lineages showed that *LRRC15+* cells were enriched in arthritis, skin wound, fibrosis, and small and large pancreatic ductal adenocarcinoma, and these cells also showed high expression of *CTHRC1* (Buechler et al, 2021). Unlike CTHRC1, LRRC15 is a cell surface marker, and an antibody drug conjugate against LRRC15 is currently under clinical development to target cancer stromal cells (Purcell et al, 2018). It will be interesting to evaluate whether it also works in the anti-fibrotic setting. Senescent cells have emerged as another prominent type of pathological cells that accumulate and contribute to fibrosis progression (Schafer et al, 2017). In future studies, it will be interesting to compare the effects of ablating *Cthrc1+* cells and p21+/p16+ (senescent) cells to determine whether an anti-fibrotic or anti-senescence CAR-T offers better therapeutic benefits in fibrosis.

# Materials and Methods

## Mice

The Calico and UCSF Institutional Animal Care and Use Committee approved all animal protocols. The sequence of the *Rosa26^LSL-OVA-Luc* allele is available in the Supplemental Data 1. *Cthrc1-CreERT2* has been reported previously (Tsukui et al, 2020). *Rosa26^TdTomato* (Ai14, Jax#007914) (Madisen et al, 2010), *Albumin-Cre* (Jax #003574) (Postic et al, 1999), *actin-Cre* (Jax #033984) (Lewandoski et al, 1997), RIP-mOVA (Jax #005431) (Kurts et al, 1996) were obtained from Jax.

Mice used in this study were between 6 and 20 wk of age. Both male and female mice were used. Mice in the same colony, not necessarily littermates, were randomly allocated for experiments. Mice that were moribund or that lost 20% of their body weight were euthanized.

## Genotyping assays for *Rosa26^LSL-OVA-Luc/+*

Primers: WT forward: GGG AGT GTT GCA ATA CCT TTC T, WT reverse: CTT TAA GCC TGC CCA GAA GA, mutant Forward: ATC ATG TCT GGA TCC CCA TC, mutant reverse: AGG CTG AAC CGG GTA TAT GA.
WT mice: 155 bp.
Homozygous mutants: 218 bp.
Heterozygotes: 155 and 218 bp.

## Luciferase imaging in vivo

Luciferin was injected at 150 mg/kg per mouse via IP injection, and the images were taken 10–20 min postinjection.

## MEFs

MEFs were generated at Jax using a standard procedure. Briefly, female mice were placed in mating with a male for ~16 h and checked for the presence of a plug. E13.5 embryos were collected and genotyped. P0 cells were grown to confluency and split to produce the P1 population, which were further grown to confluency before being frozen down. The P1 cells were recovered and used for experiments.

## MEF and T cell co-culture assays

Spleens from wild-type C57BL/6 mice or OT-1 Rag2$^{-/-}$ mice (#2334; Taconic) were harvested, mechanistically dissociated, and filtered by a 70-μm nylon mesh filter. Red blood cells were lysed in ACK lysis buffer (A1049201; Thermo Fisher Scientific). CD8 T cells were then purified via negative selection, using MACS CD8a+ T Cell Isolation Kit (# 130-104-075; Miltenyi Biotec), following its standard protocol.

The purified T cells were cultured in RPMI 1640 media (11875-093; Fischer), supplemented with 10% fetal bovine serum (F2442-100ML; Sigma-Aldrich), 1% Penicillin–Streptomycin (1514-122; Gibco), 2 mM GlutaMax (100×, 35050-061; Fischer), 1 mM pyruvate (11360-070; Fischer), and 50 μM 2-Mercaptoethanol (M3148-25ML; Sigma-Aldrich).

T cells were resuspended at $1 \times 10^6$ cells/ml in the culture media, and activated by CD3/CD28 Dynabeads (25 μl for $1 \times 10^6$ cells in 1 ml; Invitrogen), in the presence of 50 IU/ml IL-2 (#212-12; PeproTech). Dynabeads were removed when cells were split the first time after activation (48 h after activation). These pre-activated T cells (3–4 d after activation) were then co-cultured with GFP or Cre lentivirus infected MEFs for 24 h, and the cell-killing efficiency was assessed by the change of luminescence signals.

## In vivo ablation of OVA+ cells

For T adoptive transfer approach: spleens of wild-type C57BL/6 mice, or OT-1 Rag2$^{-/-}$ mice (#2334; Taconic) were obtained, and spleens of the same genotype were pulled together for CD8$^+$ T cell purification, using the MACS CD8a+ T Cell Isolation Kit. 5–10 million OT-1 or wild-type T cells were then resuspended in PBS and injected into recipient mice via tail vein injection, with the maximum volume not exceeding 200 μl/mouse.

For OVA antibodies: 10 mg/kg antibodies were dosed via intraperitoneal injection twice per week.

To measure blood glucose level, a glucometer was used, using drops of blood collected via tail bleeding. To measure liver enzyme levels, ~100 μl of the serum was collected via retro-orbital bleeding, and those samples were analyzed in a serum chemistry analyzer (Beckman Coulter).

## Generation of the OVA antibodies

The variable region of anti-OVA antibody was generated by mouse immunization with chicken OVA. B cells of immunized mice were isolated by single-cell sorting. cDNA for variable regions (both heavy and light chains) was amplified by reverse transcription PCR and cloned into expression vectors containing DNA sequences for either mouse IgG1 or mouse IgG2 constant regions. Both variants of the antibody were expressed in HEK293 cells, purified by Protein A chromatography, and confirmed to bind chicken OVA in ELISA.

## ADCC reporter assay

The ADCC activity of anti-OVA antibodies were measured with mouse FcγRIV ADCC Bioassay kit (M1201; Promega). Target cells were plated in 96-well white flat bottom plates together with anti-OVA mIgG1, anti-OVA mIgG2a or mIgG2a isotype control at indicated concentrations, and mFcγRIV effector cells at a 3:1 effector: target cell ratio. After 6 h of incubation at 37°C, Bio-Glo Luciferase Assay Reagent was added and luminescence was determined with CLARIOstar (BMG LABTECH).

## Cell surface antibody binding assay

Single-cell clones of NIH3T3 cells were blocked in PBS containing 1% goat serum and incubated with titrated anti-OVA mIgG2a, anti-OVAmIgG1 or mIgG2a isotype antibodies for 30 min at 4°C. Cells were washed three times with PBS containing 1% BSA and incubated with Alexa Fluor 647 anti-mIgG (115-605-062; Jackson ImmunoResearch) and LIVE/DEAD Fixable Violet Dead Cell Stain (L34964; Thermo Fisher Scientific). The cells were washed twice and FACS data were collected via the BD FACSCanto II HTS. The A647 Geo means gated live singlets were used to generate the graphs. FlowJo and GraphPad Prism were used for data analysis.

## Bleomycin-induced lung fibrosis model

The model was described previously in Tsukui et al (2020). Briefly, for a T cell-based approach, mice were treated with 2 U/kg bleomycin in 70 µl saline by oropharyngeal aspiration to induce fibrosis at Day 0. Tamoxifen was administered at 2 mg/mouse from Day 8–12 and Day 15–19. Mouse T cells were isolated from spleens and lymph nodes of C57BL/6 OT-1 mice (C57BL/6-Tg(TcraTcrb)1100Mjb/J, Strain #003831 from Jackson Laboratories), which were mechanically dissociated over a 40 micron filter. RBCs were lysed using RBC lysis buffer (Biolegend) before negative selection for CD3+ T cells (StemCell) with purity confirmed post-sort by surface staining. Mouse T cells were grown in RPMI supplemented with 10% fetal bovine serum, 2 mM Glutamax, 20 mM HEPES, 1% pen/strep, 1 mM sodium pyruvate, 0.05 mM β-mercaptoethanol, and 50 IU/ml human IL-2. Mouse T cells were activated on the day of isolation with OVA peptide (GenScript). 24 h after activation, $1 \times 10^6$ mouse T cells were spinfected at 2,000$g$ for 2 h at 32°C on retronectin-coated (15 µg/ml; Takara Bio) non-TC-coated 24 well plates with 1 ml of retrovirus and 4 ng/ml polybrene (Sigma-Aldrich). Retrovirus was removed after 4 h and mouse T cells expanded in OVA until 2 d after activation. Mouse T cells were then expanded daily by counting and

diluting with mTCM + IL-2 to maintain a concentration of $1 \times 10^6$ cells/ml for 9 d after activation before use with in vivo assays.

Mice received 150 mg/kg cytoxan by i.p. injection 1 d before adoptive transfer. 3 million GFP-labeled WT or OT-1 T cells were transferred via tail vein on day 14. Lungs were harvested on day 21. Left lobes were used for hydroxyproline assay. Right lobes were dissociated and used for flow cytometry. The other lobes were used for histological analysis. For antibody-based approaches, mice were treated with 1 U/kg bleomycin in 70 µl saline by oropharyngeal aspiration at Day 0 to induce fibrosis. Tamoxifen was administered at Day 7–11 and Day 14–18 at 80 mg/kg daily, and antibodies were dosed via IP injection twice per week (Days 4, 7, 10, 14, 17) at 10 mg/kg.

## Hydroxyproline quantification content

Fibrosis after bleomycin treatment was assessed by hydroxyproline assay of tissue lysates as described previously (Tsukui & Sheppard, 2022 Preprint). Briefly, left lobes were homogenized and precipitated with trichloroacetic acid. After baking at 110°C overnight in HCl, samples were reconstituted in water, and hydroxyproline content was measured by a colorimetric chloramine T assay.

## Lung sample flow cytometry analysis

Lungs were dissociated as previously described (Tsukui et al, 2020). Briefly, mouse lungs were harvested after perfusion through the right ventricle with PBS. The right upper lobe of the lung was collected. After mincing with scissors, the tissue was resuspended in a protease solution (0.25% Collagenase A [Millipore Sigma], 1 U/ml Dispase II [Millipore Sigma], 2000 U/ml Dnase I [Millipore Sigma] in RPMI [Millipore Sigma] supplemented with 10 mM HEPES). The suspension was incubated at 37°C for 60 min with trituration by micropipette every 20 min. Then, the cells were passed through a 100 µm cell strainer (BD Biosciences), washed with PBS, and resuspended in PBS with 0.5% BSA (Fisher BioReagents).

Those prepared single-cell suspensions were then used for flow cytometry analysis. Non-specific antibody binding was minimized by anti-CD16/CD32 antibody (#553142; BD Biosciences), and cells were stained with fluorophore-conjugated antibodies against: CD31 (390) [endothelial cells], CD45 (30F-11) [hematopoietic cells], EpCAM (G8.8) [epithelial cells], CD146 [edndothelial cells], Ter119 (TER-119) [red blood cells] to mark non-fibroblast cell lineages. The lineage-negative and tdTomato+ cells were gated as the *Cthrc1+* cells in *Cthrc1-CreERT2*; *Rosa26^{Td-tomato/LSL-OVA-Luc}* mice. FlowJo and GraphPad Prism were used for data analysis.

## Stable cell line generation

The single-cell clones of NIH3T3 cells were generated at GenScript. Briefly, the same TFRC-OVA construct used in the mouse study (sequence available in the supplement) was cloned into pCDH EF1-MCS-(PGK-Puro) (CD810A-1) backbone. Two stable pools of cells were generated using low (MOI = 1) and high (MOI = 40) MOI lentiviral infections, using DMEM, 10% FBS, and 2 µg/ml puromycin media for selection. Clones with different OVA expression levels were selected and confirmed by flow cytometry analysis, using a

commercial OVA antibody (520402; BioLegend) ([Fig S3A](link)). The cells were maintained in DMEM, 10% FBS, and 2 $\mu$g/ml puromycin.

## Immunohistochemistry

Tissue sections at 5 $\mu$m in thickness were deparaffinized in xylene, immersed in decreasing concentrations of ethanol, and rehydrated in water. H&E staining was done following the standard protocol. For IHC staining, the sections were pretreated using heat-mediated antigen retrieval with either sodium citrate buffer or Tris EDTA buffer for 15 min. After blocking endogenous peroxidase activity with 3% $H_2O_2$ in water for 10 min, the sections were further blocked with 1× casein solution containing 2% normal goat serum for 30 min at room temperature. The sections were then incubated with a primary antibody (OVA antibody; Millipore, AB1225, at 1:1500 dilution; Anti-CD4: Abcam, ab183685, Lot# GR3240246; Anti-CD8 antibody: Cell Signaling #98941) for overnight at 4°. After several rinses in TBST, the sections were incubated with the polymer detection system for 30 min. The bound peroxidase was visualized by incubating the sections with ImmPACT DAB Peroxidase (HRP) Substrate. The slides were counterstained with modified Mayer's hematoxylin and cover slipped.

For immunofluorescence imaging of tdTomato and GFP lungs were fixed with 4% paraformaldehyde, cryoprotected by 30% sucrose, and embedded in OCT compound (Sakura Finetek). 12-$\mu$m cryosections were stained with DAPI (0.1 $\mu$g/ml), mounted with Prolong Glass (Thermo Fisher Scientific), and imaged by Nikon Ti2 confocal microscopy.

## qRT-PCR

10,000 cells from single-cell suspension after lung dissociation were lysed in 400 $\mu$l Trizol (Ambion) and RNA was isolated according to the manufacturer's protocol. The RNA was reverse transcribed using a Super Script IV VILO Master Mix with ezDNase Enzyme kit (Thermo Fisher Scientific). Quantitative real-time PCR was performed using PowerUp SYBR Green Master Mix (Thermo Fisher Scientific) with a Quant Studio 4 (Applied Biosystems). We used the following primers: *Rps3* forward 5'-CGGTGCAGATTTCCAAGAAG-3' and reverse 5'-GGACTTCAACTCCA-GAGTAGCC-3', *Cthrc1* forward 5'-AAGCAAAAAGCGCTGATCC-3' and reverse 5'-CCTGCTGGTCCTTGTAGACAC-3', *Col1a1* forward 5'-AGACATGTTCAGCTTTGTGGAC-3' and reverse 5'-GCAGCTGACTT-CAGGGATG-3'.

# Supplementary Information

# Acknowledgements

We would like to thank Johannes Riegler for the help with luciferase imaging; Chirag Patel for the troubleshooting of FACS; the Laboratory Animal Resources department for the support of in vivo work; Calvin Jan and Astrid Gillich for the critical reading of the manuscript; and Cynthia Kenyon for the support of the project. This work was also supported by funding from Calico Life Sciences LLC and grants HL155786 (T Tsukui), HL142568 (D Sheppard).

## Author Contributions

J Zhang: conceptualization, data curation, formal analysis, investigation, methodology, project administration, and writing—original draft, review, and editing.
T Tsukui: data curation, formal analysis, investigation, methodology, project administration, and writing—original draft, review, and editing.
X Wu: data curation, formal analysis, investigation, and methodology.
A Brito: data curation, formal analysis, investigation, methodology, and writing—original draft.
JM Trumble: data curation, formal analysis, investigation, and methodology.
JC Caraballo: data curation, formal analysis, investigation, and methodology.
GM Allen: data curation, formal analysis, investigation, methodology, and writing—original draft.
J Zavala-Solorio: data curation, formal analysis, investigation, and methodology.
C Zhang: data curation, formal analysis, investigation, and methodology.
J Paw: data curation, formal analysis, investigation, and methodology.
WA Lim: supervision, funding acquisition, and investigation.
J Geng: supervision, funding acquisition, investigation, and writing—original draft.
Y Kutskova: supervision, funding acquisition, investigation, project administration, and writing—original draft, review, and editing.
A Freund: conceptualization, supervision, funding acquisition, investigation, and project administration.
G Kolumam: supervision, funding acquisition, investigation, and project administration.
D Sheppard: supervision, funding acquisition, investigation, project administration, and writing—review and editing.
RL Cohen: supervision, funding acquisition, investigation, and writing—original draft, review, and editing.

## Conflict of Interest Statement

All Calico and Abbvie affiliated authors were employees of the companies at the time of this study. JM Trumble is now an employee of Moderna. J Paw is now an employee of Cytek Biosciences. A Freund is now an employee of Arda Therapeutics. D Sheppard is a founder of Pliant Therapeutics and has received research funding from Abbvie, Pfizer, and Pliant Therapeutics. D Sheppard serves on the Scientific Review Board for Genentech and on the Inflammation Scientific Advisory Board for Amgen. WA Lim holds equity in Gilead and Intellia, is an adviser for Allogene Therapeutics and has filed patents related to this work.

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
