## [Reviewer comments · Life Science Alliance]

Life Science Alliance

An immune-based tool platform for in vivo cell clearance

Jieqiong Zhang, Tatsuya Tsukui, Xiumin Wu, Alyssa Brito, John Trumble, Juan Caraballo, Greg Allen, José Zavala-Solorio, Chunlian Zhang, Jonathan Paw, Wendell Lim, Jiefei Geng, Yuliya Kutskova, Adam Freund, Ganesh Kolumam, Dean Sheppard, and Robert Lawrence Cohen

DOI: <https://doi.org/10.26508/lsa.202201869>

Corresponding author(s): Robert Lawrence Cohen, Calico and Jieqiong Zhang, Calico

Review Timeline:	Submission Date:	2022-12-07
	Editorial Decision:	2023-01-26
	Revision Received:	2023-05-04
	Editorial Decision:	2023-05-24
	Revision Received:	2023-05-30
	Accepted:	2023-05-31

Scientific Editor: Novella Guidi

Transaction Report:

January 26, 2023

Re: Life Science Alliance manuscript #LSA-2022-01869-T

Dr. Robert Cohen
Calico
1170 Veterans Blvd
South San Francisco, CA 94080

Dear Dr. Cohen,

Thank you for submitting your manuscript entitled "An immune-based tool platform for in vivo cell clearance" to Life Science Alliance. The manuscript was assessed by expert reviewers, whose comments are appended to this letter. We invite you to submit a revised manuscript addressing the Reviewer comments.

Thank you for this interesting contribution to Life Science Alliance. We are looking forward to receiving your revised manuscript.

Sincerely,

B. MANUSCRIPT ORGANIZATION AND FORMATTING:

Reviewer #1 (Comments to the Authors (Required)):

Zhang et al. describe a mouse model with expression of floxed membrane-anchored ova and luciferase at the Rosa26 locus. By crossing with Cre transgenic mice they generate mice expressing ova and luciferase in specific cell types. In the generated models they evaluate targeting of these ova-expressing cells by transfer of either OT-1 cells or ova-specific antibodies. A limitation of the study is that only ova with a known strong CTL epitope targeted by the OT-1 cells is used and not a clinically more relevant target protein. Even in this setting a high effector to target cell ratio and a strong ova expression is needed for efficient targeting of cells in vitro or in vivo.

- Description of methods is rather short and some things are missing. More details or references to previously published methods would be helpful for others to replicate the study.
- Many of the figure legends miss crucial information or do not describe the figure sufficiently. For many legends the information about scale bars in histological images, whether mean or median, how many replicates or which type of error (SD/SEM) etc. is missing.
- For Figure 1b only one exemplary animal is shown per group. It would be helpful to include a luciferase quantification for all animals similar as in Figure S1c.
- For Figure S1d it is mentioned in the text that no effects were seen for the GFP control lenti. These data should be added to the graph.

Reviewer #2 (Comments to the Authors (Required)):

There is growing recognition that immune-mediated cytotoxic therapies that target pathological cells may be beneficial in a number of chronic diseases beyond the remarkable success of immune checkpoint inhibitors for cancer. In this study, the authors describe a technique that will enable labeling cells-of-interest with surface expressed antigen ovalbumin (OVA), followed by transfer of antigen-specific T cells that eliminate these cells in-vivo. They show that fibrogenic cells can be eliminated using this strategy to reduce fibrosis in the lung, and this was more effective than use of OVA antibodies.

The main concern with the paper is that fibrosis in the lung had marginal effect when targeting Cthrc1-expressing fibroblasts as evidenced by hydroxyproline (Fig. 3E), despite more impressive reduction in Col1a1 and Cthrc1 (Fig. 3D). This should be explained or reconciled experimentally.

We addressed the review comments in blue. Underscores were used to highlight the changes that we made in the manuscript.

Reviewer #1 (Comments to the Authors (Required)):

Zhang et al. describe a mouse model with expression of floxed membrane-anchored ova and luciferase at the Rosa26 locus. By crossing with Cre transgenic mice they generate mice expressing ova and luciferase in specific cell types. In the generated models they evaluate targeting of these ova-expressing cells by transfer of either OT-1 cells or ova-specific antibodies.

- A limitation of the study is that only ova with a known strong CTL epitope targeted by the OT-1 cells is used and not a clinically more relevant target protein. Even in this setting a high effector to target cell ratio and a strong ova expression is needed for efficient targeting of cells in vitro or in vivo.

Yes, we agree that the OVA platform has its limitations. We therefore modified the text, further emphasizing the limitations.

“One important caveat of the system is that OVA is exogenously introduced and is typically overexpressed at an artificially high level. Therefore, having established that a cell type is worth pursuing, it will be important to take into consideration the immunogenicity, specificity, endogenous copy number of the candidate surface markers, and number of pathological cells that need to be targeted.”

In terms of the E:T ratio, adoptive transfer of T cells only caused relatively mild liver damage in Rosa26-OVA, Alb-Cre model. This is a special case, where billions of hepatocytes are expressing the target, which makes it infeasible to achieve a high E:T ratio, not to mention the liver's robust regeneration capacity. In typical therapeutic applications, it is unlikely the relevant pathological cells would be as abundant as the hepatocytes. For example, millions of T cells, when adoptively transferred, were effective in clearing out beta cells in RIP-mOVA mice, and Cthrc1+ pro-fibrotic cells in Rosa26-OVA, Cthrc1-Cre mice. We therefore believe our model, when applied in relevant settings, can serve as a useful pilot tool.

- Description of methods is rather short and some things are missing. More details or references to previously published methods would be helpful for others to replicate the study.

Thanks for the feedback.

We added more details in the methods, covering T cell harvest, T cell culture, the adoptive transfer approach, stable cell line generation, lung sample flow cytometry analysis.

For example, T cell harvest procedure was added:

“Spleens from wild type C57/B6 mice, or OT-1 Rag2^{-/-} mice (Taconic, #2334) were harvested, mechanically dissociated, and filtered by a 70 μm nylon mesh filter. Red blood cells were lysed in ACK lysis buffer (Thermo Fisher, A1049201). CD8 T cells were then purified via negative selection, using MACS CD8a⁺ T Cell Isolation Kit (# 130-104-075, Miltenyi Biotec), following its standard protocol. The purified T cells were cultured in RPMI 1640 media (Fischer 11875-093), supplemented with 10% Fetal Bovine Serum (Sigma F2442-100ML), 1% Penicillin Streptomycin (Gibco 1514-122), 2 mM GlutaMax (100x, Fischer 35050-061), 1 mM Pyruvate (Fischer 11360-070), and 50 μM 2-Mercaptoethanol (Sigma, M3148-25ML). T cells were resuspended at 1*10⁶ cells/ml in the culture media, and activated by CD3/CD28 Dynabeads (Invitrogen, 25 ul for 1*10⁶ cells in 1ml), in the presence of 50 IU/ml IL-2 (#212-12, PeproTech). Dynabeads were removed when cells were split the first time after activation (48h after activation). These pre-activated T cells (3-4 days after activation) were then co-cultured with GFP or Cre lentivirus infected MEFs for 24 hours, and the cell killing efficiency was assessed by the change of luminescence signals.”

Additional details describing flow cytometry analysis of lung samples were added:

“Lungs were dissociated as previously described (Tsukui et al., 2020). Briefly, mouse lungs were harvested after perfusion through the right ventricle with PBS. The right upper lobe of the lung was collected. After mincing with scissors, the tissue was resuspended in protease solution [0.25% Collagenase A (Millipore Sigma), 1 U/ml Dispase II (Millipore Sigma), 2000 U/ml Dnase I (Millipore Sigma) in RPMI (Millipore Sigma) supplemented with 10 mM HEPES]. The suspension was incubated at 37 °C for 60 min with trituration by micropipette every 20 min. Then the cells were passed through 100 μm cell strainer (BD Biosciences), washed with PBS, and resuspended in PBS with 0.5% bovine serum albumin (BSA) (Fisher BioReagents). Those prepared single cell suspensions were then used for flow cytometry analysis. Non-specific antibody binding was minimized by anti-CD16/CD32 antibody (BD Biosciences, #553142), and cells were stained with fluorophore-conjugated antibodies against: CD31 (390), CD45 (30F-11), EpCAM (G8.8), CD146, Ter119 (TER-119) to mark non-fibroblast cell lineages. Those lineage negative and tdTomato⁺ cells were gated as the Cthrc1⁺ cells in Cthrc1-CreERT2; Rosa26^{Td-tomato/LSL-OVA-Luc} mice. FlowJo and GraphPad Prism were used for data analysis.”

- Many of the figure legends miss crucial information or do not describe the figure sufficiently. For many legends the information about scale bars in histological images, whether mean or median, how many replicates or which type of error (SD/SEM) etc. is missing.

Thanks for the feedback. Indeed, after reflecting on the reviewer's comment, we agree that the figure legends needed much improvement. We therefore updated all figure legends throughout the manuscript.

- For histological images, scale bars, and number of animals were specified.

For example, Figure 2C:

"H&E staining of liver histology analysis. Mice from (B) were taken down at Day 14. $n = 3$ for mice injected with mlgG1 control antibody; $n = 5$ for mice injected with mlgG2a antibody. Scale bar, 100 μm . Arrowheads: inflammation and granuloma."

- For quantification results, the indication of symbols/lines (mean or median, individual animals), the number of replicates, if individual data was not shown, the type of error, and the p value evaluation are included.

For example, Figure 2B:

"Serum liver enzyme analysis post antibody injection in $Rosa26^{\text{LSL-OVA-Luc}/+}$, $Alb\text{-Cre}$ mice. Symbols represent individual mice. Lines indicate mean. Multiple Mann-Whitney tests were used to determine p-values in panel B."

- For Figure 1b only one exemplary animal is shown per group. It would be helpful to include a luciferase quantification for all animals similar as in Figure S1c.

Thanks for the comments. We included the quantification results in Figure 1B, and the number of animals in the figure legend.

Representative luciferase imaging results of Rosa26^{LSL-OVA-Luc}+, crossed with different Cre lines. Quantification results were shown in the panel below. n=3 or 4 mice per genotype. Symbols represent individual mice. Lines indicate median.

- For Figure S1d it is mentioned in the text that no effects were seen for the GFP control lenti. These data should be added to the graph.

Thanks for the comments.

The GFP control data supporting the statement was embedded in Figure 1D. When MEFs were cultured alone without the addition of T cells, those MEFs infected with Cre, but not GFP, expressed the luciferase reporter (Figure 1D, first set of columns, when T cells/MEF ratio=0, compare GFP vs Cre infected cells).

Co-Culture killing assay

We also have additional data supporting the statement. In the early optimization experiments, MEFs harvested from Rosa26-OVA mice transfected with Cre plasmids, but not GFP, expressed the luciferase reporter (See below, data not included in the manuscript).

Figure: MEFs harvested from Rosa26-OVA mice, or WT mice were transfected with different plasmids, as indicated in the figure legend. WT MEFs infected with pLuc (plasmid containing a luciferase reporter) was included as a positive control.

To keep the manuscript concise, we adjusted the text, but did not include those early optimization results in the manuscript.

“These MEFs showed luciferase signals when infected with Cre lentivirus (Figure S1D), but not GFP (Figure 1D, first column, T cells/MEF ratio=0).”

Reviewer #2 (Comments to the Authors (Required)):

There is growing recognition that immune-mediated cytotoxic therapies that target pathological cells may be beneficial in a number of chronic diseases beyond the remarkable success of immune checkpoint inhibitors for cancer. In this study, the authors describe a technique that will enable labeling cells-of-interest with surface expressed antigen ovalbumin (OVA), followed by transfer of antigen-specific T cells that eliminate these cells in-vivo. They show that fibrogenic cells can be eliminated using this strategy to reduce fibrosis in the lung, and this was more effective than use of OVA antibodies.

The main concern with the paper is that fibrosis in the lung had marginal effect when targeting Cthrc1-expressing fibroblasts as evidenced by hydroxyproline (Fig. 3E),

despite more impressive reduction in Col1a1 and Cthrc1 (Fig. 3D). This should be explained or reconciled experimentally.

Thanks for the very constructive feedback.

From the single cell RNA-seq data that we published earlier (see below), we have identified that the Cthrc1+ cells, or those fibrosis-associated fibroblasts (Cluster 8 cells) express the highest levels of ECM proteins, evidenced by the expression of fibrotic genes Collagen 1a1 (Col1a1) and Cthrc1 in bleomycin-treated mice (Tsukui et al., 2020).

In the T cell mediated targeting experiment, we adoptively transferred T cells to clear out those fibroblasts on Day 14, and measured the hydroxyproline content and the expression of fibrotic genes at Day 21. While qPCR data were like snapshots at the time of harvesting, hydroxyproline content reflects the total collagen accumulated during fibrosis over 21 days, including the non-pathological collagen.

The reduction of Col1a1/Cthrc1 expression, together with our FACS and imaging results (Figure 3B, 3C), support the conclusion that our immune-mediated cytotoxic approach is effective in removing Cthrc1 expressing fibroblasts. In contrast, the putative therapeutic benefit measured by the hydroxyproline content is relatively mild (Figure 3E). It is likely that our approach does not inhibit early fibrogenesis, which happens even before the T cells were introduced (Day 14). This is not too surprising given the complicated factors contributing to fibrosis and, of course, an immune-mediated cytotoxic approach is just one therapeutic strategy.

We therefore modified the following paragraph.

“Adoptive transfer of GFP-labeled OT-1 T cells resulted in infiltration of GFP+ OT-1 T cells and significant reduction in the number of tdTomato+ Cthrc1+ cells (Figure 3B, 3C). qPCR analysis for whole lung cells also confirmed strong reduction of Cthrc1 and Col1a1 (Collagen 1a1) expression (Figure 3D), further supporting the efficient removal of the Cthrc1+ profibrotic cells, the main cell types contributing to the expression of fibrotic genes (Tsukui et al., 2020). In contrast, administration of the OVA antibodies for two weeks did not lead to reduction of tdTomato+ Cthrc1+ cells (Figure S4B-D).

This set of experiments showed that the OVA system can be applied to specifically label a disease-associated cell type. In this fibrosis model, T cells rather than the OVA antibody are much more effective in eliminating Cthrc1+ cells in vivo. Importantly, in mice where Cthrc1+ fibroblasts were eliminated, fibrosis measured by the hydroxyproline content was also modestly reduced (Figure 3E). Together, these results suggest while the elimination of Cthrc1 cells can be effective, the therapeutic benefits in improving fibrosis are relatively mild. It is likely that our approach does not inhibit early fibrogenesis, which happens even before the T cells were introduced (Day 14). Therefore, the marginal therapeutic benefits are not too surprising given the complicated factors contributing to fibrosis and, of course, an immune-mediated cytotoxic approach is just one therapeutic strategy.”

May 24, 2023

RE: Life Science Alliance Manuscript #LSA-2022-01869-TR

Dr. Robert Lawrence Cohen
Calico
1170 Veterans Blvd
South San Francisco 94010

Dear Dr. Cohen,

Thank you for submitting your revised manuscript entitled "An immune-based tool platform for in vivo cell clearance". We would be happy to publish your paper in Life Science Alliance pending final revisions necessary to meet our formatting guidelines.

- please add ORCID ID for the secondary corresponding author-they should have received instructions on how to do so
- please add keywords for your manuscript to our system
- please add the Twitter handle of your host institute/organization as well as your own or/and one of the authors in our system
- please make sure that the author order in your manuscript and the author order entered in our system match and that every author's name is entered in our system correctly
- please consult our manuscript preparation guidelines <https://www.life-science-alliance.org/manuscript-prep> and make sure your manuscript sections are in the correct order
- please use the [10 author names, et al.] format in your references (i.e. limit the author names to the first 10)
- please double-check your figure callouts for Figure S2; you have a callout for Figure S2B but this panel isn't in the legend or the figure
- please add a callout for Figure S4A to your main manuscript text

A. FINAL FILES:

B. MANUSCRIPT ORGANIZATION AND FORMATTING:

Sincerely,

Reviewer #1 (Comments to the Authors (Required)):

Thank you for addressing all my comments

May 31, 2023

RE: Life Science Alliance Manuscript #LSA-2022-01869-TRR

Dr. Robert Lawrence Cohen
Calico
1170 Veterans Blvd
South San Francisco 94010

Dear Dr. Cohen,

Thank you for submitting your Research Article entitled "An immune-based tool platform for in vivo cell clearance". It is a pleasure to let you know that your manuscript is now accepted for publication in Life Science Alliance. Congratulations on this interesting work.

DISTRIBUTION OF MATERIALS:

Again, congratulations on a very nice paper. I hope you found the review process to be constructive and are pleased with how the manuscript was handled editorially. We look forward to future exciting submissions from your lab.

Sincerely,
